

# Influence of allelic variations in relation to norepinephrine and mineralocorticoid receptors on psychopathic traits: a pilot study

Guillaume Durand

Department of Psychiatry & Neuropsychology, University of Maastricht, Netherlands

## ABSTRACT

**Background**. Past findings support a relationship between abnormalities in the amygdala and the presence of psychopathic traits. Among other genes and biomarkers relevant to the amygdala, norepinephrine and mineralocorticoid receptors might both play a role in psychopathy due to their association with traits peripheral to psychopathy. The purpose is to examine if allelic variations in single nucleotide polymorphisms related to norepinephrine and mineralocorticoid receptors play a role in the display of psychopathic traits and executive functions.

**Methods**. Fifty-seven healthy participants from the community provided a saliva sample for SNP sampling of rs5522 and rs5569. Participants then completed the Psychopathic Personality Inventory–Short Form (PPI-SF) and the Tower of Hanoi.

**Results**. Allelic variations of both rs5522 and rs5569 were significant when compared to PPI-SF total score and the fearless dominance component of the PPI-SF. A significant result was also obtained between rs5522 and the number of moves needed to complete the 5-disk Tower of Hanoi.

**Conclusion**. This pilot study offers preliminary results regarding the effect of allelic variations in SNPs related to norepinephrine and mineralocorticoid receptors on the presence of psychopathic traits. Suggestions are provided to enhance the reliability and validity of a larger-scale study.

## INTRODUCTION

Psychopathy is commonly defined as a personality condition characterized by an absence of emotional empathy, impulsivity, callousness, and manipulative behaviors (*Berg et al., 2013*; *Gao & Tang, 2013*; *López et al., 2013*). Although the causes of psychopathy are unknown, research supports a combined influence of genetic, environmental, and developmental factors (*Berg et al., 2013*). A meta-analysis of twin studies in the field of psychopathy supports a considerable (29–56%) genetic influence on psychopathic traits (*Rhee & Waldman, 2002*). While there is an ongoing debate regarding the brain structures the most relevant in psychopathy, previous findings support the role of a paralimbic system dysfunction as a central component of psychopathy (*Blair, 2007*; *Hyde et al., 2014*;

Corresponding author
Guillaume Durand,
gdura061@uottawa.ca

*Yoder, Porges & Decety, 2015*). A literature review identified implications between psychopathy and orbital frontal cortex, insula, anterior and posterior cingulated, parahippocampal gyrus, anterior superior temporal gyrus, and amygdala (*Kiehl, 2006*). Among these structures, there is a relative consensus that a dysfunctional amygdala (i.e., a lack of activation in the amygdala during fMRI scans in psychopathic individuals on tasks related to psychopathy, such as empathy) plays a major role in psychopathy (*Blair & Mitchell, 2009*). In their review, *Blair & Mitchell (2009)* highlight that the theory of a dysfunctional amygdala is supported by data indicating reduced emotional attention in psychopathy. Furthermore, the literature suggests that psychopathy is associated with numerous core functional impairments, such as deficits in aversive conditioning, augmentation of the startle reflex by visual threat primes and fearful expression recognition (*Blair, 2007*). These impairments are also seen following lesions of the amygdala (*Blair, 2006*). However, since psychopathy is not a neurological condition, nor is it associated with cerebral lesions, other factors, such as biomarkers, may play a role in the expression of psychopathy.

Various biomarkers have been associated with the display of psychopathic traits, such as cortisol and testosterone (*Glenn, 2009*). Out of these biomarkers, norepinephrine (NE) has received considerable attention due to its role in emotional processing of the amygdala (*Chrousos & Gold, 1992*; *McGaugh, 2000*). A past study investigated the effect of betablocker (propranolol; a noradrenergic antagonist), using highly emotional stimuli (*Van Stegeren et al., 2005*). The authors monitored amygdala activation with fMRI during encoding of sets of pictures between participants on the betablocker and those on the placebo. The findings support the role of NE in amygdala activation. Indeed, neutral and very light emotional pictures did not activate the amygdala significantly compared to the baseline level, while negative emotional pictures resulted to a significant increase in amygdala activation, but only under the placebo condition. When both central and peripheral noradrenergic receptors were 'blocked' using propranolol, amygdala activation decreased when participants were presented emotional stimuli. These results go in line with previous findings linking NE and aggressive behaviors (*Craig & Halton, 2009*). Indeed, beta-type noradrenergic receptor blockers have also been used to control aggressive behavior in violent individuals (*Yudofsky, Silver & Hales, 1998*). These findings suggest that genetic variation in the NE receptors may be important in aggression responses. Additionally, previous reports suggest that antisocial individuals have lower baseline levels of NE, which is also a hallmark of increased aggression (*Perez, 2012*). Altogether, these findings indicate a potential relationship between psychopathic traits and dysfunctional NE secretion.

In addition to NE, the mineralocorticoid receptor (MR) might play a central role in psychopathy due to its effect in peripheral traits of psychopathy (i.e., traits that may not be present in every psychopathic individuals, such as boldness and fearlessness, as opposed to core traits which constitute the hallmark of psychopathy, such as disinhibition and meanness) (*Lynam et al., 2011*; *Ter Heegde, De Rijk & Vinkers, 2015*; *Venables, Hall & Patrick, 2014*). Previous findings support an association between MR and risk taking (*Deuter et al., 2017*), MR and stress resilience (*Ising et al., 2008*), and MR and moderation of childhood emotional neglect and amygdala reactivity (*Bogdan, Williamson & Hariri, 2012*).

Although it is highly debated as to if risk taking and heightened levels of stress resilience are core features of psychopathy or components peripheral to the condition (*Benning, 2013*; *Blonigen, 2013*; *Lilienfeld, 2013*; *Marcus, Edens & Fulton, 2013*; *Patrick, Venables & Drislane, 2013*), these traits are nonetheless associated with psychopathy. Indeed, past research identified a relationship between taking risky decision and psychopathic traits (*Takahashi et al., 2014*), as well as stress resilience and increase in psychopathic traits (*Dunlop et al., 2011*; *Durand & Plata, 2017*; *Uzieblo et al., 2010*). Alternatively, multiple studies reported an association between childhood emotional neglect and the presence of psychopathic traits later in life (*Graham et al., 2012*; *Watts et al., 2017*). While a negative correlation is observed between psychopathic traits related to boldness and self-reported childhood maltreatment, a positive association is observed between traits related to meanness and disinhibition and childhood maltreatment. Based on the interaction between MR genotype and a history of childhood maltreatment, whereas a positive association between emotional neglect and threat-related amygdala reactivity is only observed in iso homozygotes, MR may play a role in moderating or predicting psychopathic traits.

In addition to a potential role between MR and psychopathy, MR may be associated with executive functions. Indeed, MR is present in numerous areas such as the dorsal hippocampus, the ventral hippocampus, and the medial prefrontal cortex. Furthermore MR has been previously implicated in fear and memory (*McEown & Treit, 2011*). A decrease of MR mRNA expression in the prefrontal cortex of schizophrenia and bipolar disorder has also been observed (*Xing et al., 2004*). Considering the role of the prefrontal cortex in psychopathy, which is mostly due to the ventromedial and anterior cingulated sectors, theorized to mediate numerous social and affective decision-making functions, it is possible that there is an interaction between MR, executive functions, and psychopathy (*Koenigs, 2012*). However, previous findings have obtained contradictory results regarding the type of association between executive functions and psychopathy. One study concluded that psychopaths who had never been convicted for a crime performed better than psychopaths who have previously been convicted and non-psychopathic individuals on the Wisconsin Card Sorting Test (WCST) (*Ishikawa et al., 2001*). A second study concluded that higher levels of psychopathic traits were negatively correlated with various executive functions, such as inhibition, working memory, and planning (*Lantrip et al., 2016*). These findings were however moderated when examining the results by psychopathic subtypes, whereas traits related to fearless dominance were correlated with better executive functions, while traits related to antisocial and impulsivity were correlated with worse executive functions. Another study reported that highly psychopathic individuals performed similarly to low and middle psychopathy groups on a manual version of the Tower of Hanoi, while performing better than those two groups on a computerized version of the task, which requires working memory, planning, and inhibition (*Salnaitis et al., 2011*). While the interaction effect between psychopathy groups and modality of the task is unclear, the results provide insights regarding a potential relationship between psychopathic individuals and planning abilities. Considering the supposed relationship between NE and MR on psychopathy, and the link between psychopathy and cognitive functions, it might also be possible to observe a role of NE or MR on cognitive abilities.

To the author's knowledge, no study has explicitly investigated the relationship between NE and MR from a genetic point of view, and their relationship with psychopathic traits and executive functioning. Genetic studies being increasingly expensive, a preliminary study in a healthy sample is needed to confirm the expected relationship between the aforementioned variables before engaging in a costly and time consuming larger scale study. Hence, the purpose of this pilot study is to establish a paradigm for a future study assessing the role of genetic variations in relation to NE and MR in the presence of psychopathic traits and cognitive abilities. I first hypothesize a correlation between psychopathic traits and cognitive abilities. I also hypothesize a difference between NE and MR SNPs and psychopathic traits. Lastly, I hypothesize a difference between NE and MR SNPs and cognitive abilities. To this end, two single nucleotide polymorphisms (SNP) were selected: the rs5522 of the MR gene NR3C2, and the rs5569 from the SLC6A2, which is a NE transporter. Rs5522 was selected due to its past association with enhanced physiological stress response and reduced cortisol-induced MR gene expression, two components which can be associated to psychopathy (*Bogdan et al., 2010*). Alternatively, rs5569 was selected as it is one of the most common SNP studied when examining the SLC6A2 gene (*Bruxel et al., 2014*; *Miguita et al., 2006*; *Retz et al., 2008*; *Sun et al., 2008*).

## METHODS

### Participants

A total of 57 healthy participants (Males = 30, Females = 27) were recruited via advertisements on social media, on university campus, and on site at a firefighter department to take part in the current study. DNA was collected using saliva sampling at the day of assessments. The age of the participants ranged between 18 to 59 years old ($M = 34.51$, $SD = 14.91$). Participants were recruited from universities ($N = 25$), fire departments ($N = 12$), and online community groups ($N = 20$). All participants were free of any psychotropic medication for the past 12 months. No participant reported receiving treatment from a health professional for the past six months, nor did any participant report a psychiatric or medical diagnosis. The descriptive characteristics of the participants are provided in Table 1. The current study was approved by the Ethics Review Committee of IntegReview (Austin, TX; http://www.integreview.com; protocol number 11122015). All participants received and signed an informed consent form with detailed information about the nature, the goal, the procedure and possible consequences of the study prior starting the experiment. All participants' information was kept anonymous during the courses of the whole experimental process and analyses. Participants received $10 as compensation for their time.

### Measurements

#### Psychopathic Personality Inventory–Short Form

The Psychopathic Personality Inventory–Short Form (PPI-SF) is a 56-item self-report questionnaire derived from the original 187-item PPI (*Lilienfeld & Widows, 2005*). The PPI-SF assesses psychopathic traits through eight subscales, namely: Machiavellian Egocentricity, Social Potency, Fearlessness, Coldheartedness, Impulsive Nonconformity,

**Table 1 Demographic characteristics of participants and mean score.**

| | |
|---|---|
| Sex (M/F) ($N = 57$) | (30/27) |
| Age, years: (Mean ± SD) ($N = 57$) | (34.5 ± 14.9) |
| Tower of Hanoi | |
|     3-disk tower ($N = 57$) (Mean ± SD) | (9.1 ± 4.5) |
|     4-disk tower ($N = 48$) (Mean ± SD) | (22.7 ± 9.2) |
|     5-disk tower ($N = 45$) (Mean ± SD) | (51.4 ± 17.1) |
| Completion of the 5-disk tower | 45 (79%) |
| Time to completion of the 5-disk tower in seconds ($N = 45$) (Mean ± SD) | (168.8 ± 67.8) |
| Psychopathic Personality Inventory–Short Form ($N = 57$) | |
|     PPI-SF Total | (121.7 ± 12.3) |
|     PPI-I | (52.8 ± 8.2) |
|     PPI-II | (54.1 ± 7.6) |

**Notes.**

Due to the algorithm of the Tower of Hanoi, participants must complete the 3-disk tower and the 4-disk tower in order to complete the 5-disk tower. Due to the time limitation of 5 min, nine participants were not able to complete the 4-disk tower, and 12 participants (including those of the 4-disk tower) were not able to complete the 5-disk tower. An explanatory video, showing the method to complete the 3-disk, 4-disk, and 5-disk towers is available as a Data S1.

Blame Externalization, Carefree Nonplanfulness, and Stress Immunity. The questionnaire is rated on a Likert scale ranging from 1 = *False* to 4 = *True*. Seven of the eight subscales are divided into two factors, namely Fearless Dominance (PPI-I) and Impulsive Antisociality (PPI-II). While Coldheartedness does not load on neither of these factors, it is included in the total score. PPI-I focuses on adaptive traits (social poise, fearlessness, stress immunity), while PPI-II focuses on maladaptive traits (manipulative tendencies, callousness, lack of empathy, impulsivity). A higher score correspond to higher levels of psychopathic traits.

### The Tower of Hanoi

The Tower of Hanoi is a problem-solving task measuring executive functioning. Although principally focusing on planning abilities, the task also requires working memory, inhibition, problem solving, and goal-directed behavior (*Salnaitis et al., 2011*). The goal of the puzzle is to displace a set of disks, five in the current study, from the middle rod to the first or third rod, while following three rules: (1) never move two disks at the same time, (2) only move the upper disk of the stack and (3) never place a bigger disk on a smaller disk. A previous study has determined that this puzzle act on the participant capacity to plan ahead, which is correlated with the frontal lobe functioning (*Goel & Grafman, 1995*). The task is over once the participant has successfully completed the puzzle, or after 5 min. The algorithm of the puzzle forces participants to complete a tower of three disks and a tower of 4 disks before completing the tower of five disks. The minimum number of moves to complete the 3-disk tower is 7, 15 for the 4-disk tower and 31 for the 5-disk tower. Scoring is performed by calculating the number of stack completed, the number of moves needed to complete each stack, and the time needed to complete the 5-disk tower among individuals who completed it.

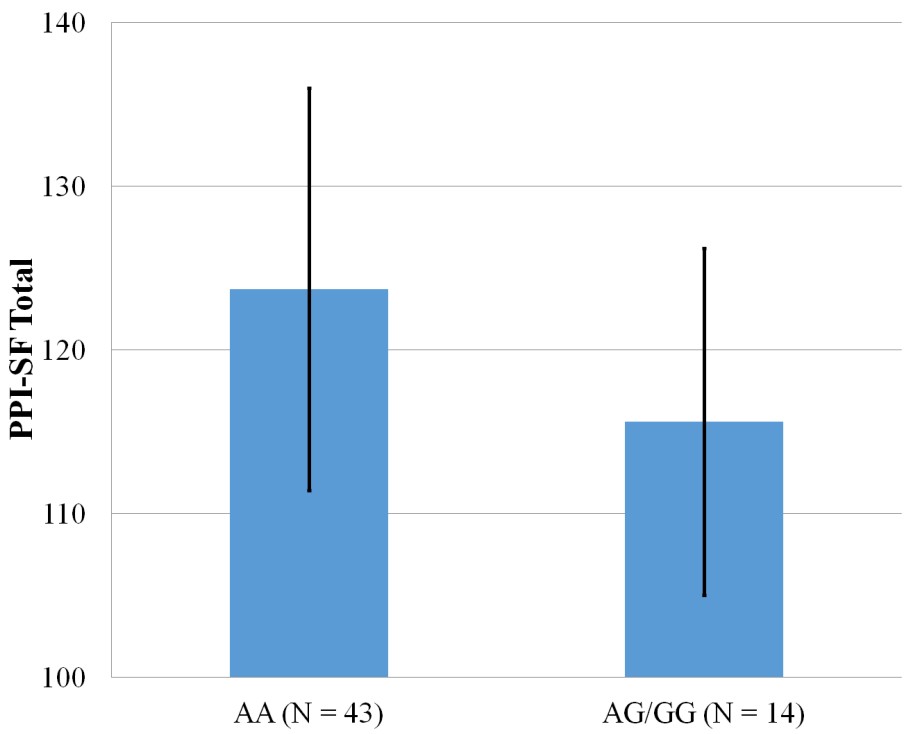

**Figure 1** Mean score on the PPI-SF Total, with error bars showing a 95% confidence interval on rs5522 genotype.

## Sample DNA extraction

The selected SNPs were processed via a Sequenom panel for multiplex reaction and genotyping at McGill University (McGill University and Génome Québec Innovation Centre, Québec, Canada). DNA Extraction was done using the prepIT-L2P from DNA Genotek (item #PT-L2P-45) according to the manufacturer's protocol. A multiplex PCR was performed on 20 ng of template genomic DNA in a 5 μL reaction mixture containing: 0.1 μL (0.5 U) HotStar Taq enzyme (QIAGEN), 0.625 uL of 10X HotStar Buffer, 0.325 μL of 25 mM (total) MgCl2, 0.25 μL of 10 mM dNTP mix, 0.55 μL of forward and reverse primer pool (1 μM) and 1.15 μL of water. The amplification cycling used was: 95c 15 min, 45× (95c 20 s, 56c 30 s, 72c 60 s), 72c 3 min, hold 4c. A few PCR reactions were run on QIAxcel (QIAGEN, Valencia, CA, USA) to assess the amplification (1 uL of PCR in 9 μL of DNA Dilution Buffer (QIAGEN, Valencia, CA, USA)). This was followed by a shrimp-alkaline-phosphatase treatment to render the leftover nucleotides unusable (0.2 μL of SAP Buffer, 0.3 μL of SAP and 1.5 μL of water). SAP cycling: 37c 40 min, 85c 10 min, hold 4c. Next, a primer extension reaction (iPLEX Gold) was performed with 0.94 μL of extension primer mix, 0.2 μL of iPLEX Terminator, 0.2 μL of iPLEX Buffer, 0.041 uL of iPLEX Thermo Sequenase and 0.619 μL of water. The products were desalted using 6 mg of resin (Agena Bioscience, San Diego, CA, USA) and spotted on a 384-point SpectroCHIP (Agena Bioscience, San Diego, CA, USA) using a nanodispenser. The distinct masses were

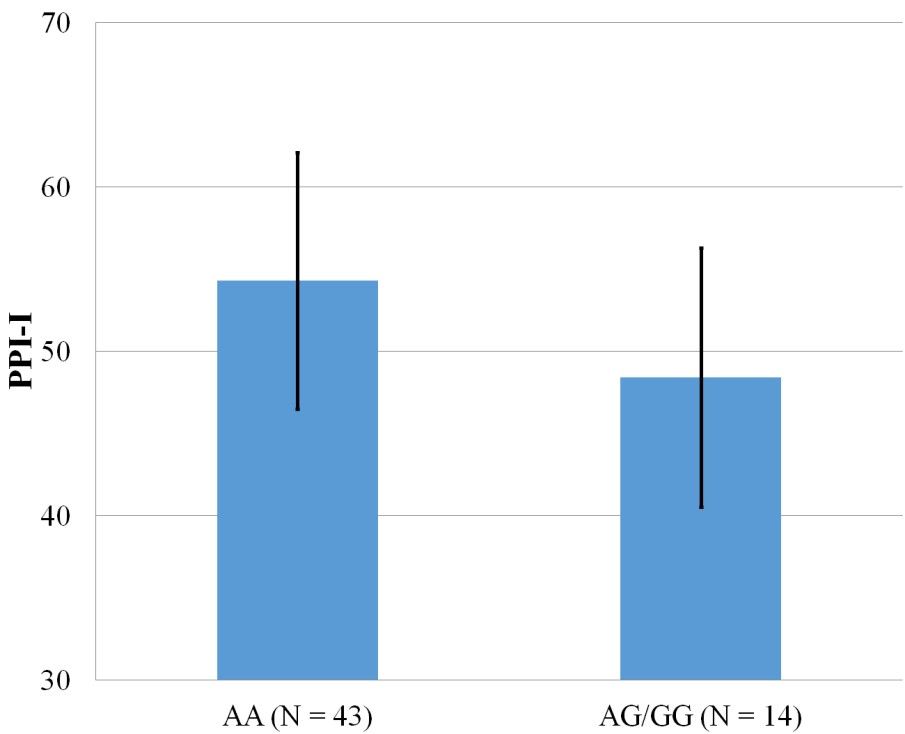

**Figure 2** **Mean score on the PPI-I, with error bars showing a 95% confidence interval on rs5522 geno-type.**

determined by MALDI-TOF mass-spectrometry and data was analyzed using MassARRAY Typer Analyser software.

## Experimental procedure

Upon the candidates' arrival to the laboratory, the participants were asked to sign the consent form. The candidates were asked to answer demographic information and a saliva sample was collected. The participants then completed the PPI-SF. Upon completion of the questionnaire, the participants completed the Tower of Hanoi.

## Statistical analysis

All analyses were performed using the Statistical Package SPSS version 23.00 (IBM Corporation, Armonk, NY, USA). Identification of dominant and recessive alleles for the two SNPs was performed using refSNP (http://www.ncbi.nlm.nih.gov/snp/). A dominant model was used for both rs5522 (AA = 1, Ag or gg = 0) and rs5569 (CC = 1, Ct or tt = 0). A series of ANOVA were performed on both SNPs and the various dependent variables.

## RESULTS

### Psychopathic traits and planning abilities

A Pearson correlation between each component of the PPI-SF (PPI-SF Total, PPI-I, and PPI-II) and the variables associated with the Tower of Hanoi (number of moves to complete

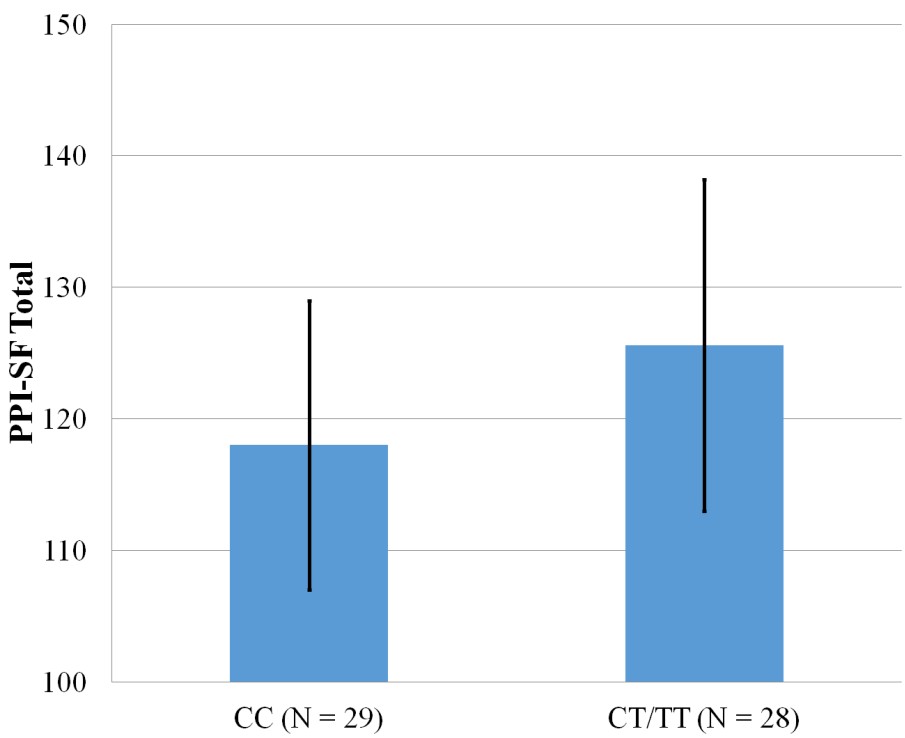

**Figure 3** Mean score on the PPI-SF Total, with error bars showing a 95% confidence interval on rs5569 genotype.

each towers and time to complete the 5-disk tower of Hanoi among those who completed it) did not yield any significant result.

### SNPs and psychopathic traits

As shown in Figs. 1–4, two significant differences were observed on both SNPs. On rs5522, AA alleles carriers displayed higher scores on PPI-SF Total ($M = 123.72$, SD $= 12.29$) than AG/GG alleles carriers ($M = 115.57$, SD $= 10.57$) ($F(1, 56) = 4.946$, $p = .030$). Additionally, AA alleles carriers also displayed higher scores on PPI-I ($M = 54.27$, SD $= 7.84$) than AG/GG alleles carriers ($M = 48.42$, SD $= 7.89$) ($F(1, 56) = 5.855$, $p = .019$). On rs5569, CC alleles carriers displayed lower scores on PPI-SF Total ($M = 118.00$, SD $= 11.03$) than CT/TT alleles carriers ($M = 125.57$, SD $= 12.57$) ($F(1, 56) = 5.845$, $p = .019$). Lastly, CC alleles carriers also displayed lower scores on PPI-I ($M = 49.75$, SD $= 8.01$) than CT/TT alleles carriers ($M = 56.03$, SD $= 7.19$) ($F(1, 56) = 9.659$, $p = .003$).

### SNPs and planning abilities

As shown in Fig. 5, only one significant result emerged from a series of ANOVA comparing the two SNPs with scores obtained on the Tower of Hanoi. Within participants who completed the last level of the Tower of Hanoi, AA carriers from rs5522 completed the puzzle in less moves ($M = 48.38$, SD $= 13.62$) than AG/GG alleles carriers ($M = 60.82$, SD $= 23.35$) ($F(1, 44) = 4.772$, $p = .034$).

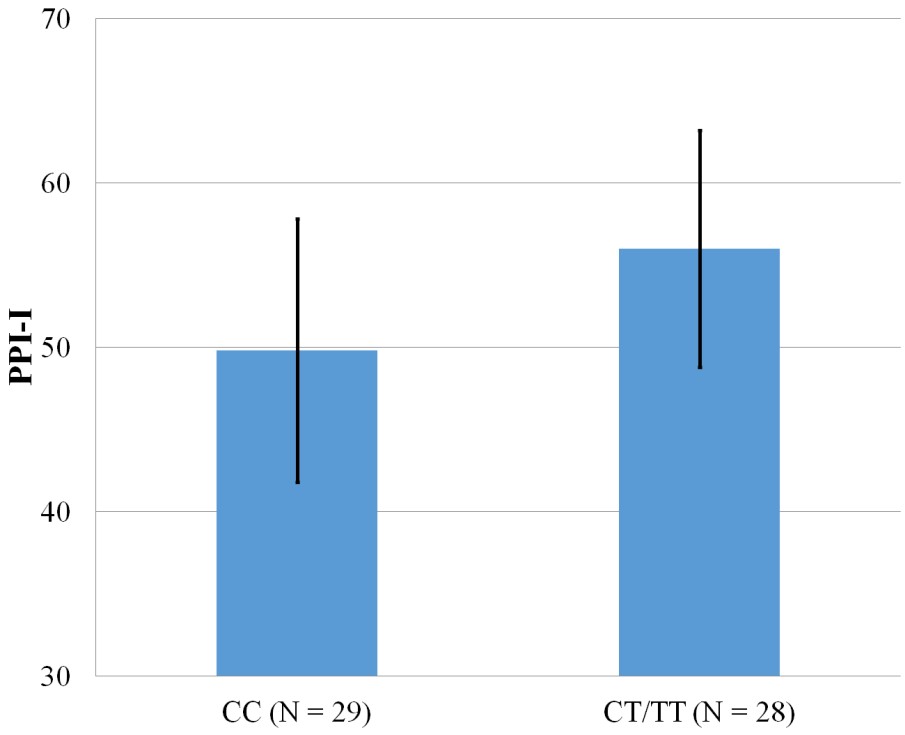

**Figure 4** Mean score on the PPI-I, with error bars showing a 95% confidence interval on **rs5569** genotype.

## DISCUSSION

This study examined the relationship between the allelic variations in two SNPs related to NE and MR in relation to psychopathic traits and executive functioning. Preliminary support was found for a relationship between both SNPs on PPI-SF total scores and PPI-I, as well as for an association between MR and planning abilities as assessed by the Tower of Hanoi.

Several conclusions can be drawn from the results of this pilot study. First, despite the low number of participants, most results were well under the threshold of $p < .05$ to establish significance. However, while the results were particularly encouraging regarding rs5569 due to its two groups including almost 30 participants each, and its particularly low $p$ value of .003 for PPI-I, the results regarding rs5522 should be taken with cautions, mostly due to the low number of participants in the non-dominant group ($N = 14$). Second, while multiple significant results were obtained regarding psychopathic traits, only one significant result emerged for executive functions. While the difference in rs5522 on the number of moves needed to complete the Tower of Hanoi was significant at $p = .034$, the non-dominant group only had 11 participants, which is fairly low to provide solid assumptions. Considering no other results emerged between the two SNPs and the Tower of Hanoi, the results indicate that a future study should provide an alternative method to assess executive functions. These results are in line with a previous study, supporting

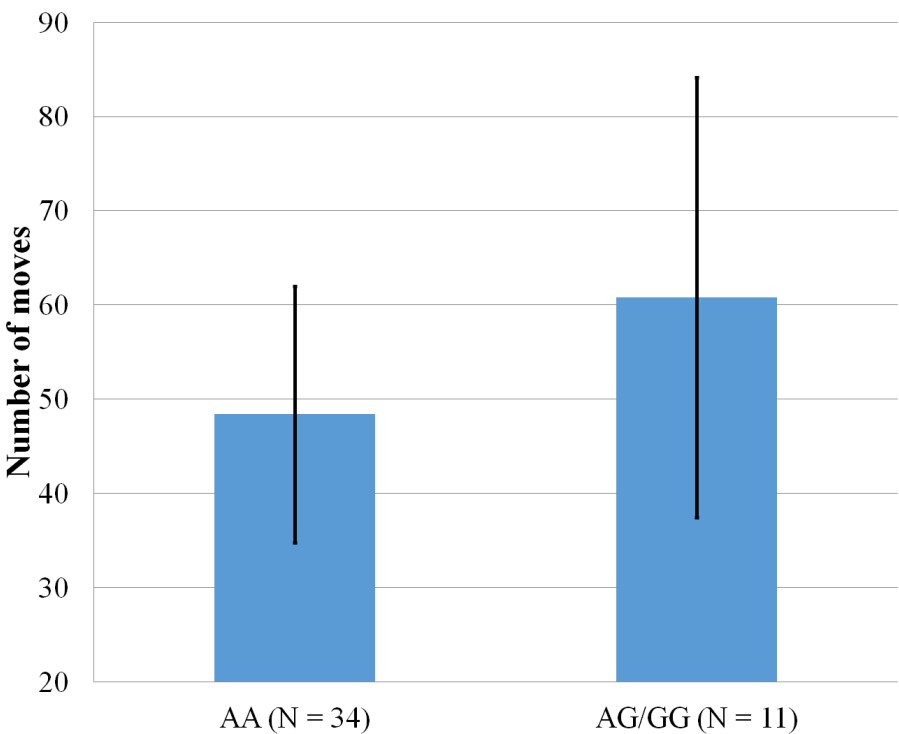

**Figure 5** Mean number of moves to complete the Tower of Hanoi, within participants who completed the 5-disk tower, with error bars showing a 95% confidence interval on **rs5522** genotype.

that the effect of psychopathic traits and scores on the Tower of Hanoi was significant for the computerized version of the task, but not on the manual version (*Salnaitis et al., 2011*). However, the results between psychopathic traits and executive functions were inconclusive. Despite the expected association between higher psychopathic traits and better results at the Tower of Hanoi, no significant association was observed.

This pilot study possesses several limitations. First, the low numbers of participant created uneven groups for rs5522. Second, only one cognitive task, namely the Tower of Hanoi, was used in the present study. Considering the failure to provide adequate results, the future study should focus on an alternative task to measure planning abilities, as well as additional tasks to measure other aspects of executive functions, such as short term memory and behavioral inhibition. Third, the participant pool is largely heterogeneous, with three distinct groups. Although a combination of students, firefighters, and adults browsing the web can arguably be more representative of the community as opposed to a single homogenous group of students, focusing on a single group might provide better results.

## CONCLUSION

Based upon these results, several aspects of this experimental design should be modified for future research. First, while the PPI-SF is a valid alternative to the long version of the PPI, it remains a shorten version of the original instrument. A complete version of an

instrument assessing psychopathic traits, comparable in size to the PPI-SF, might provides better results. For instance, the Triarchic Psychopathy Measure (TriPM; *Patrick, 2010*), assesses psychopathic in three different components, namely Boldness, Disinhibition, and Meanness, which correlates to PPI-I, PPI-II, and Coldheartedness respectively. Past findings further support the incremental validity of the TriPM over the PPI-SF (*Stanley, Wygant & Sellbom, 2013*). Second, the Tower of Hanoi did not provide the expected results. Due to the small effect size anticipated between allelic variations and executive functions, a complete battery of cognitive testing might be necessary to obtain further data regarding the relationship between those constructs. Third, I solely examined one SNP per area of interest, namely NE and MR. Although those two SNPs were significant, examining additional SNP, in addition to those examined in the present study, known to be related to NE and MR might further strengthen the conclusion of a relationship between psychopathic traits and NE and MR. Fourth, additional questionnaires should be used in conjunction to tests measuring psychopathic traits, such as measures of depressive behavior and anxiety. This would help determine if the results can be explained by other variables than psychopathic traits. These modifications to the experimental design should be sufficient for a large-scale study assessing the genetic variability and the display of psychopathic traits, in relation to executive functioning.

### Funding
The author received no funding for this work.

### Competing Interests
The author declares there are no competing interests.

### Author Contributions
- Guillaume Durand conceived and designed the experiments, performed the experiments, analyzed the data, contributed reagents/materials/analysis tools, prepared figures and/or tables, authored or reviewed drafts of the paper, approved the final draft.

### Human Ethics
The following information was supplied relating to ethical approvals (i.e., approving body and any reference numbers):

The current study was approved by the Ethics Review Committee of IntegReview (Austin, TX; http://www.integreview.com; protocol number 11122015).

### Data Availability
The data and the supplementary video are available at: Durand G. 2018. Psychopathy and genetic. Retrieved from https://osf.io/wzvay/.

## Supplemental Information

Supplemental information for this article can be found online at http://dx.doi.org/10.7717/peerj.4528#supplemental-information.

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
