# Peer review of "Influence of allelic variations in relation to norepinephrine and mineralocorticoid receptors on psychopathic traits: a pilot study"

_PeerJ, doi:10.7717/peerj.4528_

## Round 0.1 · original submission · Major Revisions

Dear Authors,The three peer reviewers have raised comments to the manuscript for your team to edit to increase the manuscript's quality and citability.

Please do the necessary revisions and resubmit back to PeerJ.

Reviewer 1 ·

Basic reporting

The present research article sought to study PPI-SF scores stratified by allelic variations in SNPs related to the NE receptor and MR.

The study seems well conducted. Nevertheless, I want to make some suggestions.

Experimental design

The study design is well defined and meaningful. Previous literature on this topic is sparse.

However, reading through the manuscript I want to make some suggestions:

1.) Introduction:
The introduction is well structured. However, I would also include some hippocampus literature when introducing the MR to the reader.

2.) Methods;
The methods are clearly explained.

3.) Results:
3a) Table 1 needs revision:
Why is table 1 showing 3 groups? This is not relevant for the main results, and I would suggest to simply make an overview of the total sample or show a table 1 stratified by the investigated allelic variations and present differences in means between the groups.
Furthermore, the first row of table 1 shows a total N = 47, which is not the case, since it should be a total N = 57.
Moreover, the authors should clarify what the Tower of Hanoi (3, 4, 5 disk tower) variables indicate (mean number of trials?).

3b.) Figure 1a+1b and Figure 2:
The N's in Figure 1+2 are confusing.

Figure 1a-d shows N = 57.
Figure 2 only shows N = 45.

Why is the N in figure 2 smaller? Please clarify what happened to the missing subjects.

The authors should mention, that their groups have different sample sizes in figure 1a+b (N=43 vs. N =14) and figure 2 (34 vs N=11). This should be considered when choosing the statistical test and during the interpretation the results.

4.) The findings are interesting, but can these findings also be (better be) explained by other psychological measures? Did the authors control for any other behavioral measures, such as depressive behavior (BDI?), anxiety (BAI?), or other scales?

Validity of the findings

Overall, I would recommend to revise the points mentioned above.

Reviewer 2 ·

Basic reporting

Line 49: consider clarifying what is meant by “dysfunctional amygdala”

Line 53, 54: the authors might want to consider discussing why the relationship between NE and alpha amylase is relevant in this context

Line 56-59: beta blockers do not block all the effects of NE. For clarity, it would be important to discuss why beta adrenergic receptors are important here, and the role of alpha adrenergic pathways. Also, avoid using blanket statements such as “inhibited the functioning of the amygdala.” Please clarify what is meant and which aspect of amygdala functioning is inhibited.

Line 60-61: The authors might want to contrast these findings with the role of NE in aggression (which in turn is associated with psychopathy). In general, aggression is associated with low levels of NE but high CSF NE has also been described as a predictor of violence in some studies.

Line 64: consider discussing MR in the prefrontal cortex where it is also expressed given the executive function changes in psychopathy

Line 65: It is unclear what “peripheral traits of psychopathy” really means

Line 76: The connection between childhood emotional neglect and psychopathy needs to be clarified further

Line 80: please explain how it is mitigated

Experimental design

Line 100-103: please describe how and why these 2 SNPs were selected

Line 107: why did the authors recruit at a fire department?

Line 134-146: the authors should consider what this test measures

Methods section: the PPI is a measure of psychopathy traits. Examining its results as a continuum in a population without ASPD is not validated. As such, the interpretation of how they relate to genetic variability can be faulty.

Validity of the findings

Results section: the connection between psychopathy and executive abilities is unclear. Overall, it appears that the relationship between the two measures and the genetic variability is missing.

The discussion needs to be redesigned with the above described concerns addressed. Otherwise, the conclusions present significant limitations.

Additional comments

General comments:
- In general, it is preferable to use DSM5 terminology. While there are certainly phenomenologic differences between psychopathy and sociopathy, I recommend using the overarching umbrella term Antisocial Personality disorder. This would still allow for further breakdown into psychopathy traits (which carry a lot of historical/social/political/legal weight)

Specific comments:
Line 45: consider adding developmental to the causes of psychopathy: “genetic, environmental [and developmental] factors”

·

Basic reporting

There are some grammatical errors in the manuscript but overall the language is sound.

The background evidence presented is of decent quality but it is not well interpreted, as I don not understand what "peripheral traits" in psychopathy are and they are not explained.

Also, the author does not explain what the connection between resilience and psychopathy is. Is it that psychopaths are more or less resilient?

The author speaks of "successful psychopaths" and "unsuccessful psychopaths" being different but does not explain what that means or how it is really relevant in this study.

I reviewed the article that they cite on The Tower of Hanoi and psychopathic tendencies (DOI: 10.1080/09084282.2010.52338). The article looks at Psychopathic traits and compares the performance on the computerized version vs. the manual version. They conclude that low/moderate psychopathy scorers performed poorly on the computerized Tower of Hanoi while high psychopathy scorers performed better on the computerized version in comparison. The conclusion was that high scores on computerized Tower of Hanoi tests should be interpreted with some caution as a result, and was not really related to the psychopathy traits the way the authors of this paper seem to imply.

Experimental design

The data surrounding NE and MR and psychopathy is nebulous at best. It was difficult to understand what the hypothesis they were testing was.
Beta Blockers do not "block" NE- they decrease response to it by competing for the receptors. This is an important distinction because some beta blockers have partial agonist activity which modulates response to NE/E and also changes the cellular response to the hormones.
As for MR receptors, it's true they're expressed in the amygdala, but what about the cortex and the prefrontal cortex? If they're commenting on executive functioning and correlating it psychopathy using the Tower of Hanoi test, then it would make sense to discuss the role of the prefrontal cortex/cortex in executive functioning and planning.
The SNPs that were chosen: why those in particular? One of them is the transporter for NE and one of them is just part of the MR gene. I don't really understand why those are relevant.
As for the use of the PPI-SF, for the most part that methodology was sound, as this scale was normed on college students. I do not particularly understand why they chose to use firefighters, however. Was there a particular type of trait in firefighters that they were looking for?

Validity of the findings

Although I am unsure of why they chose these SNPs, the processing of the samples is sound. The use of the PPI-SF is proper as this test is only normed on normal subjects and not psychopaths or jail populations.
The Tower of Hanoi test was administered properly.
The issue I have is there are a lot of jumps in logic that are not well explained by the author. He makes leaps in his conclusions and manipulates the data in a questionable manner in order to get some sort of correlation between the SNPs, the results on the PPI-SF, and the Tower of Hanoi test.
The data itself, in its raw form, may someday prove valuable, the same way genetic mapping with schizophrenics, provides databases of material that may someday unlock the mystery of the illness.
That said I do not really understand the author's conclusions which are based on arbitrary alleles (or they are not explained well enough so they seem arbitrarily chosen to the reader of the article), performance measures on the Tower of Hanoi test that measure executive functioning and ability to plan, and the PPI-SF which measures psychopathic traits in a normal population.
Coupled with a small sample size, the conclusions require a leap of faith to accept and I feel that they assume causation when they barely meet the standard of correlation.

Additional comments

The data is valuable because all data collected, especially when you're doing genetic work, can be valuable in the future. The interpretation of the results obtained is questionable at best, and is stretching the conclusions further than I think they can truly go.

---

## Round 0.2 · accepted · Accept

I am happy to convey to you and your team good news that your revised manuscript is accepted for publication in Peer J and will undergo further processing for the proofs to be prepared.

Reviewer 1 ·

Basic reporting

The authors have edited their article based on the suggestions I have made. They improved their manuscript, and clarified the paragraphs I have pointed out in my first review. For a more detailed review, please read my prior review.

Hence, I support to publish this article.

Experimental design

see above / prior review.

Validity of the findings

see above / prior review.

Additional comments

see above / prior review.

·

Basic reporting

I want to commend the author on cleaning up the language used. The manuscript flows better and reads much clearer.
The references are also properly used to explain the hypothesis and the background sets the stage in a much more effective manner.
The figures are simplified nicely and more relevant.
The hypotheses are clearly stated and the results are much more relevantly presented.

Experimental design

The research question is simplified and the experimental design used remains sound.

Validity of the findings

The results as stated currently are much more valid and useful to the reader. I am impressed with the author taking reviewers' feedback into consideration and the modifications have made the article significantly better.

Additional comments

Revisions are well done. Manuscript much clearer.